# Tumour Necrosis Factor-α, Chemokines, and Leukocyte Infiltrate Are Biomarkers for Pathology in the Brains of Venezuelan Equine Encephalitis (VEEV)-Infected Mice

**DOI:** 10.3390/v15061307

**Published:** 2023-05-31

**Authors:** Amanda L. Phelps, Francisco J. Salguero, Laura Hunter, Alexander L. Stoll, Dominic C. Jenner, Lyn M. O’Brien, E. Diane Williamson, M. Stephen Lever, Thomas R. Laws

**Affiliations:** 1Defence Science and Technology Laboratory, Salisbury SP4 0JQ, UK; 2UK Health Security Agency, Salisbury SP4 0JG, UK

**Keywords:** alphavirus, VEEV, venezuelan equine encephalitis virus, inflammation, cytokines, chemokines, leukocytes, pathology, mouse

## Abstract

Venezuelan equine encephalitis virus (VEEV) is a disease typically confined to South and Central America, whereby human disease is characterised by a transient systemic infection and occasionally severe encephalitis, which is associated with lethality. Using an established mouse model of VEEV infection, the encephalitic aspects of the disease were analysed to identify biomarkers associated with inflammation. Sequential sampling of lethally challenged mice (infected subcutaneously) confirmed a rapid onset systemic infection with subsequent spread to the brain within 24 h of the challenge. Changes in inflammatory biomarkers (TNF-α, CCL-2, and CCL-5) and CD45^+^ cell counts were found to correlate strongly to pathology (R>0.9) and present previously unproven biomarkers for disease severity in the model, more so than viral titre. The greatest level of pathology was observed within the olfactory bulb and midbrain/thalamus. The virus was distributed throughout the brain/encephalon, often in areas not associated with pathology. The principal component analysis identified five principal factors across two independent experiments, with the first two describing almost half of the data: (1) confirmation of a systemic Th1-biased inflammatory response to VEEV infection, and (2) a clear correlation between specific inflammation of the brain and clinical signs of disease. Targeting strongly associated biomarkers of deleterious inflammation may ameliorate or even eliminate the encephalitic syndrome of this disease.

## 1. Introduction

Venezuelan equine encephalitis virus (VEEV) is a single-stranded positive-sense encephalitic alphavirus capable of causing severe disease in humans and equines. During a large outbreak in Venezuela and Colombia in 1995, approximately 75,000–100,000 people were affected; neurological impairment was reported in ~4% of cases, and the case fatality rate was <1% [1]. Symptoms in people range from a mild febrile illness to acute encephalitis when exposed via an infected mosquito bite, with a potentially increased case fatality rate when exposed inhalationally after accidental laboratory exposure [2]. VEEV is considered a biological warfare agent [3] for which there are currently no licensed medical countermeasures, although an investigational new drug (IND) vaccine is available for particularly vulnerable groups, such as laboratory workers [4,5]. In recent times, significant efforts have been and are being made along the pathway to licensure of new-generation vaccines [6,7,8,9], with phase I clinical trials underway [10,11]. However, vaccination is typically used as a preventative measure, and strategies that protect against the established disease are highly desirable. Research in the field of post-exposure medical countermeasures against VEEV (and other encephalitic alphaviruses) has predominantly been focused on small molecule inhibitors and monoclonal antibodies [12,13,14,15,16,17,18,19,20,21,22,23], and these have had varying degrees of efficacy and remain unlicensed. The immune system can also present a point of medical intervention where strategies that both increase antiviral activity and/or reduce deleterious inflammation might be considered. One such strategy is PEGylated interferon alpha, which has been shown to have efficacy against VEEV in a lethal mouse model [24]. Additionally, commercially available off-the-shelf (COTS) drugs known to target inflammation for conditions such as arthritis have also been shown to be effective against VEEV in vitro [25], an approach termed ‘repurposing’. Repurposing has been demonstrably effective during the COVID-19 pandemic, where multiple COTS drugs have been shown to be beneficial to patients under, for example, the RECOVERY trial [26]. Repurposing COTS drugs to mitigate the considerable inflammatory response to VEEV infection may ameliorate or even eliminate encephalitis and seizures.

The nature of the damaging inflammation caused by encephalitic alphavirus infection is, in part, characterised. Melatonin [27] and angiotensin receptor [28] have beneficial effects through immune-regulatory functions in a mouse model of VEEV infection, and blockade of IL-1β was found to be profoundly beneficial in a mouse model of neuro-adapted Sindbis virus, another member of the alphavirus genus [29]. Evidence suggests that inflammation induced by VEEV infection is Th1-biased with a significant component of tumour necrosis factor-α (TNF-α), interferon-γ, interleukin-1 (IL-1), and IL-6 [30,31,32]. Furthermore, increasing evidence demonstrates that key forms of inflammation are detrimental during infection. Mouse knockouts in the TNF receptor (TNFR) and inducible nitric oxide synthesis (iNOS) were, in part, protected from infection with VEEV [33]. Moreover, complete protection was afforded to Toll-like receptor (TLR)-4 knockout mice, despite significant viral titres in the brain. In these TLR-4 knockout mice, the inflammatory response and permeability of the blood–brain barrier (BBB) were significantly dampened compared to wild-type control mice [34]. Imaging techniques have led to similar findings in VEEV-infected mice, identifying time-dependent increases in brain inflammation and apoptosis, as well as a reduction in BBB integrity [35]. Additionally, severe combined immunodeficiency (SCID) mice do not succumb to VEEV infection at the same rate as immune-competent mice, also suggesting that inflammation is the major contributor to morbidity, rather than any immune deficiency [36].

In the subcutaneous mouse model of VEEV infection, viral replication occurs first in dendritic cells, where it is then efficiently transported to local draining lymph nodes, with active viraemia evident by 12 h post-infection [37]. Systemic infection ensues, with particular tropism for lymphoid tissues in mice. Mice are able to clear VEEV from lymphoid tissues, blood, and other tissues. However, infection of the brain is already significant at this time. Within 6 to 24 h post-infection, the virus is present in the brain, and between 24 and 48 h post-infection, the BBB is significantly compromised [30]. The BBB is a biological feature with the primary function of protecting the central nervous system (CNS) [38]. The loss of BBB function may also provide access to pharmaceutical intervention.

Further characterisation of alphavirus-induced inflammation of the brain may inform opportunities for clinical intervention, such as repurposing COTS drugs to mitigate the considerable inflammatory response to encephalitic alphavirus infection. Here, we investigate inflammation in our mouse model specifically to find biomarkers for the pathology. These biomarkers can then be used as a more easily measured surrogate for brain disease in future studies.

## 2. Materials and Methods

### 2.1. Cell Lines and Virus

Vero cells (ATCC CCL-81) from the European Culture Collection of Animal Cell Cultures (ECACC, Salisbury, UK) were maintained in Dulbecco’s minimal essential media (DMEM) supplemented with 2 mM L-glutamine, 100 U/mL penicillin, 100 μg/mL streptomycin, and 10% (v/v) foetal calf serum (FCS), at 37 °C in a 5% (v/v) CO_2_ humidified atmosphere. For viral infections, cells were maintained in Leibovitz L-15 medium supplemented with 2 mM L-glutamine, 100 U/mL penicillin, 100 μg/mL streptomycin, and 2% (v/v) FCS at 37 °C (without CO_2_) in a humidified atmosphere. (All reagents purchased from ThermoFisher, Loughborough, UK).

VEEV serogroup IA/B, Trinidad donkey (TrD), was originally provided by Dr R Shope of the University of Texas Arbovirus Research Unit, TX, USA. Stocks were prepared by inoculating suckling mouse pups intra-cranially with approximately 500 pfu in 10 uL, incubating for approximately 24 h, and harvesting the brain tissue into L-15 medium supplemented with 2% (v/v) FCS, 2 mM l-glutamine, 50 IU/mL penicillin, and 50 µg/mL streptomycin. This was then passed through a 70 µm nylon cell strainer (Corning Falcon, ThermoFisher, Loughborough, UK), clarified at 10,000 rpm for 10 min in an SW28 rotor (Beckman Coulter, High Wycombe, UK), and stored at −80 °C. This production method minimises the potential for loss of virulence factors because of cell culture adaptation, and provides a representative wild-type virus population (quasispecies). All work with VEEV TrD was performed under ACDP (Advisory Committee on Dangerous Pathogens, UK), Containment Level 3 (CL3) conditions unless the virus had been inactivated with formaldehyde.

### 2.2. In Vivo Studies

All procedures involving animals were conducted under a Project Licence approved by internal ethical review and the UK Home Office, and in accordance with both the Animal (Scientific Procedures) Act (1986), the 1989 Codes of Practice for the Housing and Care of Animals used in Scientific Procedures and approved Animal Care and Use Review Office appendices. Groups of age-matched (6 to 8 weeks old) female Balb/c mice were obtained from Charles River Laboratories (UK) and were randomised into cages, and cages were randomly assigned into groups. They were housed on a 12 h day–night light cycle, with food and water available ad libitum in a CL3 rigid-wall isolator, complying with British standard 5726 and the 2000 European standard EN 12469. Prior to entering study conditions, mice were allowed to acclimatise for a minimum of 5 days.

A lethal dose of VEEV TrD was prepared by serial dilution in L-15 medium supplemented with 2 mM L-glutamine, and kept on ice until use. Virus was administered by the subcutaneous route (100 µL/mouse), with challenge preparations back-titrated to confirm dose received. Following challenge, a series of five pre-determined cull points were selected to represent key stages of disease, encompassing baseline (no disease), early-stage infection, peak systemic disease, through to encephalitis, and late-stage disease. All mice were weighed daily and clinically scored at least twice a day, increasing to 4-hourly at the onset of severe clinical signs (in accordance with UK Home Office requirements). Clinical scoring of mice was blinded, and scores were assigned on a scale of 0 (absent), 1 (observable), 2 (moderate), and 3 (pronounced), specifically focusing on coat condition, body posture, respiratory state, eye condition, and activity/behaviour. Any mouse observed to have a pronounced activity/behaviour score (e.g., unable to reach food and water), pronounced and very laboured respiratory rate, signs of neurological abnormality such as persistent circling, consistent head tilting or limb paralysis, or to have lost 30% of their original body weight on 2 consecutive days (or ≥33% on any single occasion), was immediately culled on welfare grounds. All culls were performed using a UK Schedule 1 method (cervical dislocation followed by confirmation of cessation of heartbeat).

Two independent experiments were conducted to enable refinements (if necessary) and a reduction in number of animals (if possible) between the studies, whilst ensuring an overall appropriately powered analysis. The two studies were identified as Experiment 1 and Experiment 2. Observations from the first study (Experiment 1) did not allow for a reduction in the number of animals for Experiment 2, and no refinements were identified.

Experiment 1: A total of 45 mice were challenged with 23 pfu/100 µL VEEV TrD subcutaneously. Of note were three mice ultimately excluded from any analysis; one cull required immediately after challenge, and two mice that were found to have not received the full challenge dose (noted and observed in line with all other animals; both were observed to be free of clinical signs of disease or weight loss throughout). At each pre-determined cull point (days 0, 1, 3, 4, and 6 post-challenge), up to ten mice were terminally anaesthetised using gaseous halothane in a bell jar and bled by cardiac puncture. Following confirmation of death, the heads of five mice were immediately taken and placed in neutral buffered formalin prior to histopathological processing. The other five mice were subject to excision of the spleen and brain for virological and immunological analysis. This included ten mice culled on day 0 (a no-challenge control group) to establish a baseline for each of these analysis areas. A further five animals not included in any pre-determined culls served to verify lethality in the model.

Experiment 2: A total of 45 mice were challenged with 4140 pfu/100 µL VEEV TrD subcutaneously; a higher challenge dose was delivered, requiring 4-hourly checks to be implemented from day 0. At each pre-determined cull point (days 0, 1, 2, 3, and 5 post-challenge), ten animals were processed in the same manner and frequency as in Experiment 1. Again, a further five animals not included in any pre-determined culls served to verify lethality in the model.

### 2.3. Plaque Assay

Vero cells were seeded into 24-well or 6-well plates at a density of 1–$5\;\times \;10^{5}$ cells/mL in DMEM supplemented with 2 mM L-glutamine, 100 U/mL penicillin, 100 μg/mL streptomycin, and 10% (v/v) FCS and incubated at 37 °C in a 5% (v/v) CO_2_ humidified atmosphere for 1–3 days. On the day of infection, virus was ten-fold serially diluted in Leibovitz L-15 medium supplemented with 2 mM L-glutamine, 100 U/mL penicillin, 100 μg/mL streptomycin, and 2% (v/v) FCS. The cell culture media was removed from the seeded plates, and dilutions were transferred to wells (100 µL for 24-well plate and 500 µL for 6-well plate) in either duplicate or triplicate and allowed to adsorb at room temperature for 30 min, with occasional rocking. An amount of 1 mL (for 24-well plate) or 5 mL (for 6-well plate) of carboxymethylcellulose (CMC) overlay media (3% (*w*/*v*) CMC diluted 1:1 in double-strength Leibovitz L-15 medium) was added to each well, and plates were incubated for 3 days at 37 °C in a humidified atmosphere, without CO_2_. Cells were fixed to a minimum final concentration of 1% (v/v) formaldehyde overnight and stained with 0.1% (w/v) crystal violet solution to visualise plaques for counting. The limit of detection in this assay was 10 pfu/mL for titrations performed in 24-well plates and 2 pfu/mL for titrations performed in 6-well plates.

Mouse tissues (whole brain and spleen) were homogenised through a 40 μm cell sieve (Corning Falcon, ThermoFisher, Loughborough, UK) into 1 mL of phosphate-buffered saline (PBS). Serial 10-fold dilutions were prepared (also in PBS) for a standard 24-well format plaque assay, as described above, for both blood and homogenised tissues. Neat samples were also plated. The remaining tissue suspensions were used in the immunological methods below. Blood was collected into EDTA tubes containing 10 μL of 100 mM EDTA after cardiac puncture to prevent clotting, and assayed in the same manner as for tissues, but without the need for homogenisation.

### 2.4. Immunological Methods

A200 μL aliquot of blood (collected into EDTA) was transferred to an empty micro-centrifuge tube and immediately centrifuged for 5 min at 300× *g*. The plasma was carefully removed by pipetting and stored at −80 ℃ for later cytokine analysis. Samples of unstained blood cells were pooled to provide a negative control in the later flow cytometry analysis. The red blood cells within all the pellets were then lysed by adding 1.6 mL of red cell lysis buffer, inverting multiple times, incubating at room temperature for 5 min and finally centrifuging for 5 min at 300× *g*. Lysed blood and buffer were discarded, and the pellets resuspended in 150 μL of FACS buffer (PBS with 2% foetal bovine serum) containing Fc block (2 μL of anti-CD16/32, Biolegend Cat #101320). After 20–30 min incubation at room temperature, 50 μL of master mix antibodies (Bio-legend: anti-CD45 (Fluorochrome BV711, clone 30-F11), anti-CD3 (Fluorochrome BV421, clone 17A2), anti-Ly-6G (Fluorochrome APC-Fire750, clone 1A8), anti-CD19 (Fluorochrome AF647, clone 6D5), and anti-CD11b (Fluorochrome PE, M1/70)) was added with the exception of the unstained controls. These were incubated for a further 40 min before adding 1.6 mL of FACS buffer and centrifuging for 5 min at 300× *g*. The supernatants were discarded, and the cells were resuspended in 250 μL of 4% paraformaldehyde in PBS. These were incubated for a minimum of 36 h at 4 ℃. Organ tissue suspensions were handled in the same manner as the blood, with the exception that brain tissue suspensions used 400 μL (not 200 µL) of initial suspension; the brain samples were resuspended in 500 μL of 4% paraformaldehyde, and, rather than carefully removing plasma, organ supernatant was tipped out into fresh tubes and stored for later use. Flow cytometry analysis was performed using the CyTek Aurora platform flow analyser; data were collected using SpectroFlo V2.0. Single colour controls and unstained cells were used for spectral unmixing calculations. Data were analysed using the software FlowJo V10.8.

Cytokine concentrations were measured using Multiplex Cytokine Assay kits (R&D systems, Minneapolis, MN, USA) and were performed in accordance with the manufacturer’s instructions. Plasma was diluted 1:5, and the supernatant from the organs was diluted 1:2, prior to addition to the assay. The results of the assays were read using a MagPix Luminex reader, and raw fluorescence data were exported for further analysis (see Section 2.6 below).

### 2.5. Histopathology Methods

Whole mouse heads were fixed in neutral buffered formalin for a minimum of 15 days before being decalcified for 7 days using the EDTA-based solution, Osteosoft (101728, Sigma Aldrich, Gillingham, UK). Samples were then trimmed sagittally to expose the brain/encephalon and nasal cavity, before being routinely processed into paraffin wax blocks. The 4 µm thick sections were taken on SuperfrostPlus slides and were stained with haematoxylin and eosin (H&E).

Immunohistochemistry was performed using the Leica Bond RXm and the Bond Polymer Refine Detection kit with horseradish peroxidase (HRP) (DS9800, Leica Biosystems, Wetzlar, Germany) to visualise microglia (anti-Iba1) and astrocytes (anti-GFAP). Sections were first dewaxed, rehydrated, and treated with 3–4% hydrogen peroxide to quench endogenous peroxidase activity (5 min). Sections were then pre-treated using either heat-induced epitope retrieval using Leica Bond epitope retrieval solution 1 (pH 6) (AR9961, Leica Biosystems) for 20 min for anti-Iba1 or by enzyme pre-treatment, Leica Enzyme 2 (AR9551, Leica Biosystems, Germany, Wetzlar) for 15 min for anti-GFAP. After antigen retrieval, anti-Iba1 (rabbit polyclonal, Fujifilm, WI, USA, Madison,. 019-19741) was incubated at a dilution of 1:1000 for 15 min, and anti-GFAP (Rabbit polyclonal. Dako, Santa Clara, CA, USA. Z0334) was incubated at the same dilution and time. 3,3′-Diaminobenzidine (DAB) was used as the chromogen for visualisation before the sections were counterstained using Harris’ haematoxylin for 10 min. Control sections of nasal cavity, lung and liver (anti-Iba1), and nasal cavity and brain (anti-GFAP) were also stained as comparators to ensure antibody staining was specific.

RNAscope, an in situ hybridisation method used on formalin-fixed paraffin-embedded tissues, was used to identify VEEV nucleic acid in the liver and spleen. Tissues were pre-treated with hydrogen peroxide for 10 min (room temperature), with target retrieval buffer for 15 min (98–101 °C), and with protease plus for 30 min (40 °C) (all Advanced Cell Diagnostics, Newark, NJ, USA). A specific probe to hybridise with VEEV nucleic acid (Advanced Cell Diagnostics, USA, Newark) was incubated with the tissues for 2 h at 40 °C. Amplification of the signal was carried out following the RNAscope protocol (RNAscope 2.5 HD Detection Reagent—Red) using the RNAscope 2.5 HD Red kit (Advanced Cell Diagnostics, USA).

Sections were randomised prior to examination, and a blind evaluation of samples was conducted by a qualified veterinary pathologist using both light microscopy and Hamamatsu NanoZoomer S360 digital slide scanner and viewed with NDP.view2 software (version 2.8.24). Table 1 details the quantitative scoring system established and used in these studies. Additionally, Figure 1 provides a representative longitudinal section from the head, showing the areas of the brain evaluated within the brain/encephalon (olfactory bulb, isocortex, hippocampus, midbrain/thalamus, cerebellum, pons, and the olfactory mucosa, including olfactory nerves).

### 2.6. Statistical Analysis

Raw data generated from the Multiplex Cytokine assays were analysed by taking the median fluorescence and using non-linear regression modelling in Graphpad PRISM V8.0. A three-parameter polynomial model was used where the y values were weighted (1y2).

For each animal, a mean value was calculated for the pathology scores shown in Table 1 (ranging from zero to four) derived from the seven brain areas that were considered. This was used as a surrogate for overall brain health. Some cytokine titres (calculated as described above) were below an estimation threshold. For these, the minimum value rounded down to the nearest significant figure was substituted. Graphs were prepared using the software Graphpad PRISM V8.0. All statistical analysis was performed using SPSS V27.0. Cytokine, viral titre, and cell count data were log-transformed prior to analysis to better-fit modelling assumptions. Principal component analysis (PCA) was used as an unsupervised method to find key attributes within the whole dataset. Pearson’s correlation was used to consider the relatedness of biomarkers to brain pathology at the group level. For the PCA, the data from the animals culled for histopathology were not included (as no immunological or virological data were available for these mice), and four instances of a missing data point (caused by faults with assay techniques) were inputted with the population mean for that variable.

## 3. Results

### 3.1. The Progression of VEEV Disease in Infected Mice

The Balb/c mouse model is a well-established tool for evaluating encephalitic alphavirus disease. Here, this model was utilised to assess the characteristics of inflammation induced by lethal subcutaneous infection with VEEV TrD strain. Across the two independent experiments, minor clinical signs of disease commencing 1–2 days post-challenge swiftly progressed to severe clinical signs (Figure 2A), and mice were observed to exhibit weight loss (Figure 2B), with the majority of mice succumbing to the disease within 5–6 days of challenge (Figure 2C). Severe clinical signs were typified by persistent circling behaviour, persistent tilting of the head, abnormal activity/behaviours, and an inability to move despite handling, all of which are indicative of neurological impairment and align well with previous studies [16,17,19,21]. The viral loads in the blood, brain, and spleen were also similar to those in previous studies (Figure 2D–F). Viraemia was rapid, with geometric mean titres of $1.8\;\times \;10^{5}$ and $9.5\;\times \;10^{6}$ pfu/mL 24 h post-challenge, maintaining this steady state until the final sampling point for the respective experiments, when levels reduced slightly to geometric means of $1.4\;\times \;10^{4}$ and $2.5\;\times \;10^{4}pfu/mL$ (Experiment 1 and Experiment 2). Viral loads in the spleen were similar to those found in the blood. Viral loads in the brain followed the same proliferation kinetics for the respective experiments, whereby geometric mean titres of $2.3\;\times \;10^{3}$ and $6.7\;\times \;10^{4}\;pfu/mL$ were measured 24 h post-challenge, increasing to $9.7\;\times \;10^{4}$ and $1.0\;\times \;10^{6}\;pfu/mL$ by day 3 post-challenge, culminating in high titres observed in mice that succumbed to infection. All animals in the study were humanely culled because they either had reached a pre-determined cull point for sampling or had reached a humane endpoint. In line with expectations from previous studies [39], mice that received the higher challenge dose of 4140 pfu/mouse in Experiment 2 were observed to have a slightly accelerated course of the disease, exhibiting increased viral loads in key tissues, clinical signs, and succumbing to infection approximately 24 h earlier than mice from Experiment 1 (challenged with 23 pfu/mouse). These infections were therefore found to be representative of the wider literature, have use in modelling human disease, and be able to provide samples for further analysis and exploitation.

### 3.2. Pathology and Virus Distribution in Brain/Encephalon and Nasal Cavity

Uninfected control mice did not show any significant histopathological lesions, and the presence of astrocytes and microglia was within the normal physiological range. Further, in situ hybridisation with a VEEV probe did not yield any specific labelling of uninfected control mice (Figure 3 and Figure 4). At 1 day post-challenge, only minimal changes were present, and virus RNA was not detected in any structures examined. From 3 to 6 days post-challenge, histopathological changes were evident in different brain regions, showing the highest severity in the olfactory bulb and the midbrain/thalamus, with less severity in the rest of the structures. The histopathological changes were mostly associated with neuronal death, spongiosis, and gliosis. High quantities of viral RNA were observed in the areas showing the greatest degree of pathology (olfactory bulb and midbrain/thalamus). However, viral RNA was also prevalent in other brain areas with minimal or mild histopathology: cerebellum, pons, or cortex.

A mean pathology score across brain structures was calculated to provide an estimate of the overall neuropathology in the brain over time. A clear association was observed, indicating neuropathology increases as the disease progresses over time (Figure 5). Within 1–4 days of infection, the pathology scores steadily increased, with representative mice displaying mild lesions in the olfactory bulb and midbrain (Figure 3 and Figure 4). By days 5–6, pathology scores reached a peak with evidence of marked/severe spongiosis, gliosis, and astrocytosis predominantly focused within the olfactory bulb, coinciding with the terminal stages of disease (Figure 3 and Figure 4). Neuronal death showing typical features of apoptosis was also evident at the late stages of the disease, with some evidence of glial cell death too. Cain et al. describe a similar pattern of virus dissemination and neuropathology in the brains of mice infected intranasally with VEEV TC-83, noting the same initial specific targeting of the olfactory bulb subsequently followed by dissemination into more caudal regions of the brain [40].

No significant lesions were observed in the nasal cavity (respiratory or olfactory regions). However, small amounts of viral RNA could be detected in both regions by in situ hybridisation from 3 days post-infection onwards (Figure 6A). Small amounts of viral RNA were also detected in the mandibular lymph nodes from 3 days post-infection onwards (Figure 6B). Viral RNA was not observed in the nerve terminations leading from the nasal cavity to the olfactory bulb, but could be detected within blood vessels in the most affected areas of the brain (Figure 6C).

### 3.3. Inflammatory Response to Infection

The associated inflammatory response to VEEV infection was determined by two methods. The concentrations of thirteen different inflammatory markers were estimated in the blood, brain, and spleen at predefined sampling points over the course of 5–6 days post-challenge. Flow cytometry was used to estimate the numbers and proportions of CD45^+^ leukocytes, CD19^+^ B-cells, CD3^+^ T-cells, CD11b^+^ monocytes, and Ly6G^+^ neutrophils. Too few leukocytes were found in the brains of these animals to allow meaningful estimation of the subpopulations. In total, 66 variables were measured in the animals culled for this purpose (Appendix A). An unsupervised learning technique was used to analyse these data. Principal component analysis (PCA) suggested that almost 70% of the whole dataset could be described using five principal components (factors).

The first factor explains over one-quarter of the data (Figure 7A). This component is substantially elevated from 24 h post-infection, where it gradually declines, and is slower in animals that received a lower challenge dose (Figure 7C) and does not correlate well with clinical signs of infection (Figure 7B). The data that most influenced this factor were the abundance of traditional Th1-biased inflammation markers in the blood and spleen (Figure 7D). This is representative of systemic VEEV infection and aligns well with other research [30,31,32], whereby the systemic infection is readily resolved by the host, but is the “springboard” from which the virus accesses the brain [41]. The second factor explains slightly less than 20% of the data (Figure 8A), identifying the profusion of Th1-biased inflammation markers within the brain (Figure 8B). This component increases over time (Figure 8D) and correlates well with clinical signs of infection (Figure 8C). This factor is likely to be representative of the encephalitis that is typical of lethal VEEV infection. 

Factors 3 and 4 may relate to experimental artefacts and do not associate with any part of the disease process. Factor 3 is comprised of mostly flow cytometry variables that do not change substantially over time, or correlate to disease severity (Appendix A). Factor 4 is mostly comprised of the concentration of cytokines that also do not change substantially over time, or correlate to disease severity (Appendix A. It is likely that factors 3 and 4 represent inter-experimental variation induced by the two multivariate titration techniques (flow cytometry and luminex) performed on two different occasions. For these specific variables, intra-experimental variation was not masked by substantial changes associated with infection (the Th-1-related parameters included in factors 1 and 2). Finally, factor 5 comprises flow cytometry measurements in the blood where a transient depletion of leukocytes was observed. This leukopenia is also a known feature of VEEV infection [37]. Table 2 summarises the findings of the five principal components determined using the PCA technique.

Two analytes that showed little change during infection (and were therefore observed contributing to component factor 3) were IL-4 and IL-17. IL-4 is a cytokine known to be associated with Th2-biased immune responses [42]. Such Th-2 responses are associated with anti-parasitic defences, mast cells, and eosinophils, and a lack of expression here was unsurprising. IL-17 is the key cytokine that drives a Th17 immune response, associated with some infections and autoimmune diseases [43]. IL-17 and IL-4′s notable absence, coupled with the presence of TNF-α, INF-γ, IL-6, and IL-10 in this model, indicates that the Balb/c mouse inflammatory response is strongly biased towards Th1. An increase in cytokines such as TNF-α and INF-γ is a recognised feature of VEEV infection [30], and is also recognised as BBB-disruptive agents during West Nile virus infection [44]. These observations have implications with regard to how inflammation might be targeted for therapeutic benefit.

Analysis of the flow cytometry data further supports this finding of Th1-biased response. Splenic leukocyte counts and proportions remained stable (Appendix A), whilst in the blood, there was some evidence for transient leukopenia specific to Ly6G^+^ neutrophils, CD11b^+^ monocytes and CD3^+^ T-cells (Appendix A, Figure 9), but not CD19^+^ B-cells. These cells are primarily associated with Th1 inflammation and in response to chemokines, such as those analysed here (CXCL-1, CCL-2, CCL-5).

### 3.4. Certain Inflammatory Markers Are Biomarkers for Neuropathology

The identification of correlates between pathology and other laboratory techniques can provide biomarkers for disease severity, as well as assist in understanding the mechanisms behind the disease. The mean pathology scores from Experiment 1, across groups of five mice, were used as a single data point and correlated to the mean value of measurements taken for the duplicate experiment (Experiment 2). This enables a correlation analysis across 10 data points specifically in the brain (Figure 10). Even with this small value for N, there was strong evidence for correlations between pathology and TNF-α, CCL-2, CCL-5, and CD45+ cell count in the brain; Pearson’s R = 0.9211 (0.6939 to 0.9815); R = 0.9189 (0.6864 to 0.9810); R = 0.9154 (0.6745 to 0.9801); and R = 0.9126(0.6652 to 0.9794) with 95% confidence intervals, respectively. This validates the second component of the PCA above. These observations provide a firm basis on which biomarkers might be useful to target as an inflammatory modulating strategy. With respect to the direction of signal travel, perhaps the leukocyte count is the most meaningful correlate for pathology. The leukocyte count is driven by the gradients of chemokine (the concentrations of these molecules are greater in the brain sample compared to the blood plasma, Appendix A); as they relocate to the brain, they leave a signature of MMP-9 and are activated by the available TNF-α. 

## 4. Discussion

In these studies, the mouse model of VEEV infection has been characterised in detail. The effects of the disease on the brains of mice were considered using both histological methods and immunological methods. The greatest pathology was observed in the olfactory bulb. This is consistent with the known function of this region and is therefore not paradoxical in the absence of respiratory infection and/or infection of the nasal cavity. Virions likely accessed the region across the blood–brain barrier (BBB) as free virions or via infected immune cells [45].

Unsupervised learning found disease measurements followed three key disease-associated patterns. These factors were systemic disease (that increased rapidly and then declined), brain infection (that increased consistently from infection), and transient leukopenia. Unique to this study, the immunological readouts were directly related to pathology in paired groups of mice. The greatest correlation was found in TNF-α, CCL-2, CCL-5, and leukocyte infiltrate. Correlation does offer the possibility of causation, and this creates hypotheses that need to be tested. However, TNF-α and BBB breakdown have previously been implicated in brain pathology [33,34]. Pharmacological interventions that target this process may provide a viable therapeutic option to treat this type of disease. Our data further strengthen this hypothesis.

There are multiple advantages to targeting the host response to treat the infectious disease. Firstly, drug resistance is unlikely to develop because the immune response targets and kills foreign bodies using a sophisticated and highly complex network of functions that pathogens are unlikely to evolve ways to evade. Secondly, there are a number of well-established, licensed pharmacologically active substances that target inflammation. These drugs might be repurposed from their original uses to treat unrelated inflammatory conditions. We recently proposed this repurposing strategy as an adjunctive therapy to treat the intrinsically antibiotic-resistant bacterial pathogen *Burkholderia pseudomallei* [46]. Repurposing drugs that target the host have a higher likelihood of success compared to drugs that inhibit the pathogen. This is because their pharmacodynamically active concentrations are already known to be within the pharmacokinetic limits. Drugs that have coincidental activity against a pathogen are not optimised for this purpose and may need concentrations beyond those feasibly attained in the host. More recently, this strategy of repurposing has been used with great success, typified in the RECOVERY trial to identify drugs to treat severe cases of SARS-CoV-2 disease [26] and the thousands of people who now owe their lives to treatments such as dexamethasone and tocilizumab. 

Dexamethasone was the first success of the RECOVERY trial [47], and has been in use since the 1960s to treat a range of inflammatory conditions. Dexamethasone is a glucocorticoid medication that both suppresses nuclear factor-κB and enhances mitogen-activated protein kinase activation, resulting in a general immune tempering [48]. It is, therefore, conceivable that this drug may temper the general inflammation observed during VEEV infection. The second drug identified in the RECOVERY trial was tocilizumab [49]. The target for tocilizumab is IL-6, a cytokine protein that is clearly upregulated during VEEV infection in mice and has known inflammation-regulating properties [50]. Tocilizumab has been shown to be pharmacologically active in mice and may be beneficial in treating VEEV infection [51]. Other biologics targeting the inflammatory response should also be considered. Given the high degree of correlation between neuropathology and changes in TNF-α and IL-1α levels, evaluating biologics such as etanercept and anakinra may prove useful. Moreover, targeting TNF-α pharmacologically may have similar effects to those observed in knockout mice [33,34]. Etanercept is a TNF-α binding protein that can be used clinically and (important in these studies) has been used at therapeutic doses in mice [52]. Observations in the studies described here identified only a moderate correlation of neuropathology with IL-1 and only in one form of the cytokine, IL-1α. Canakinumab is specific for IL-1β [53] and, as such, may have limited utility against VEEV infection. Anakinra targets both forms of IL-1 as a binding protein [54], and has been shown to be highly effective in protecting mice from the worse effects of *B. pseudomallei* disease [55].

Another family of drugs with immunomodulatory properties are the non-steroidal anti-inflammatory drugs (NSAIDs). These drugs interact with the cyclooxygenase (COX) proteins, instigating a suppression of prostaglandin manufacture [56]. The effect of this is that vasodilation is inhibited, and inflammation is reduced. The RECOVERY trial considered aspirin, and only the most marginal benefit was observed [57]. Aspirin (and most “classical” NSAIDs) interacts with both variants of the COX proteins (1 and 2). However, it is COX-2 that is selectively activated during chikungunya virus infection (a related non-encephalitic alphavirus) [58]. Specific inhibitors of COX-2 are in clinical use, one of which is celecoxib. Celecoxib has been shown to have an effect in vitro against VEEV infection [25,59] and may warrant further investigation. Another drug assessed in the RECOVERY trial to have a survival benefit was baricitinib [60], a small molecule that inhibits Janus kinases (JAK) 1 and 2 [61]. These kinases facilitate signal transduction from numerous inflammatory signalling receptors, such as IL-1 and TNF-α receptors [62]. This therapy has also been used in mice, and a pharmacologically active dosing regimen has been identified [63].

The impact of immune dampening and a reduction or elimination of inflammation in the brain is unlikely to rescue mice from a lethal challenge of VEEV, given the high titres and the rapid systemic spread of the virus in the brain. However, the protection may provide the crucial time needed for the host’s adaptive response to mature. Moreover, it may sufficiently widen the window of opportunity for the administration of antiviral drugs. Investigating this hypothesis is the focus of future work. The work described here provides confidence that host drug targets are active and provides the biomarkers needed to screen for activity likely to indicate an improved outcome.

## Figures and Tables

**Figure 1 viruses-15-01307-f001:**
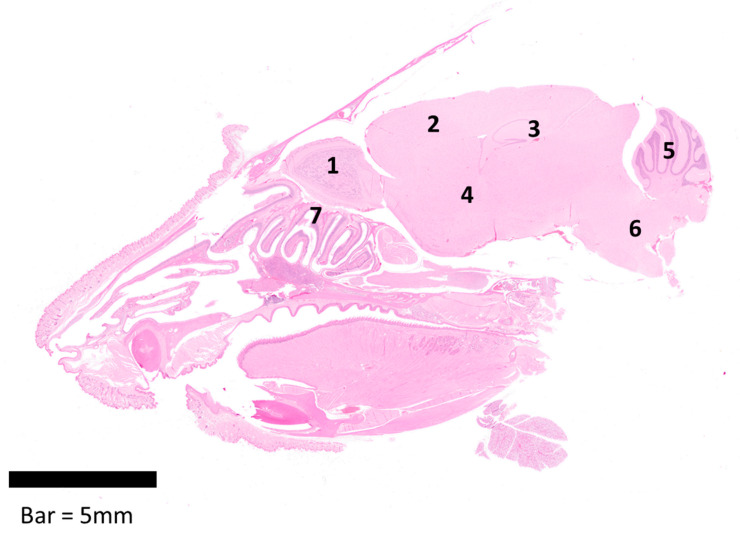
Longitudinal section of the head from a female Balb/c mouse showing the areas evaluated within the brain/encephalon (1–6) and the nasal cavity (7); 1—Olfactory bulb, 2—Isocortex, 3—Hippocampus, 4—Midbrain/Thalamus, 5—Cerebellum, 6—Pons, and 7—Olfactory mucosa, including olfactory nerves.

**Figure 2 viruses-15-01307-f002:**
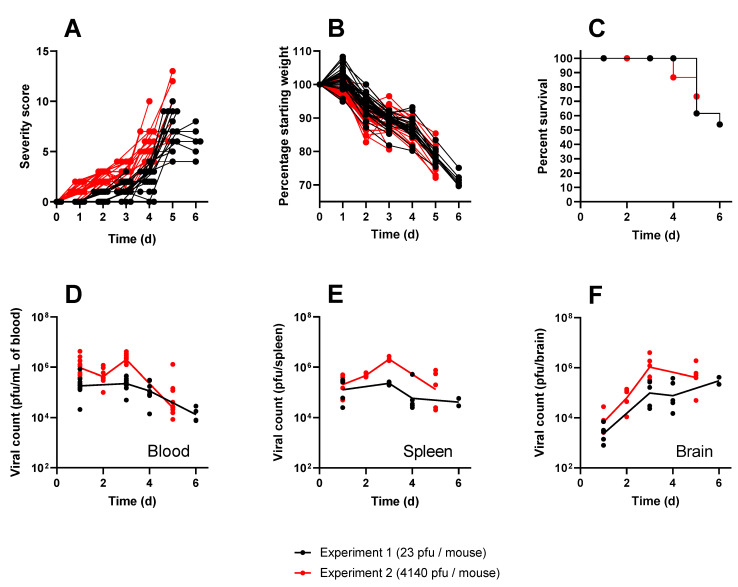
The clinical characteristics of Balb/c mice infected with either 23 pfu (black) or 4140 pfu/mouse (red) of VEEV TrD via the subcutaneous route. Panel (**A**) shows individual clinical scores from each mouse, assigned by blinded operators using pre-determined criteria. Each line represents the fate of a single mouse ending at either a pre-determined cull or lethal endpoint. Panel (**B**) shows individual weight profiles from each mouse relative to their starting weight, expressed as a percentage. Each line represents the fate of a single mouse ending at either a pre-determined cull or lethal endpoint. Panel (**C**) shows the Kaplan–Meier estimates. It should be noted that the majority of mice were culled for experimental purposes and, as regards the Kaplan–Meier estimate, are considered censored from the point of cull. For this reason, the survival in this experiment does not drop below 50%. Panels D to F show the viral titres measured at specific time points after infection in the blood (**D**), spleen (**E**), and brain (**F**). Each data point shows a single mouse, and the line is the geometric mean for each group.

**Figure 3 viruses-15-01307-f003:**
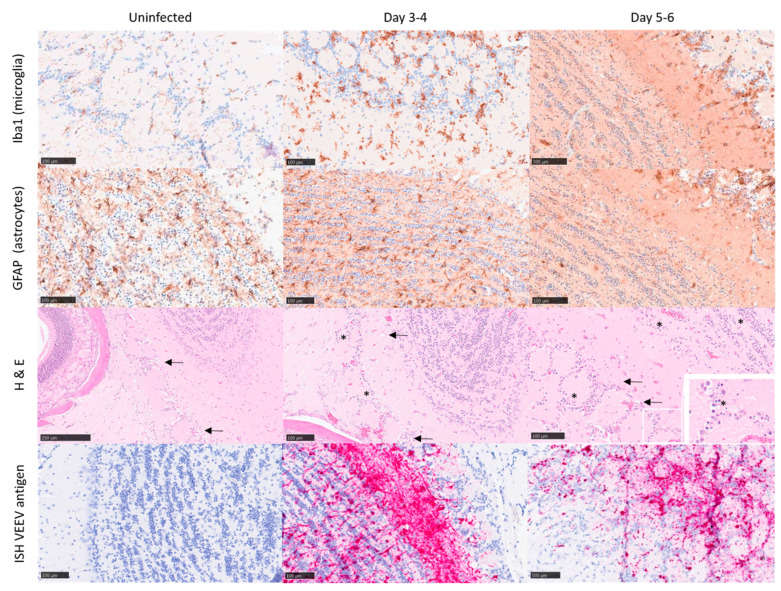
Representative images from the olfactory bulb of Balb/c mice infected with VEEV TrD subcutaneously. Uninfected controls (day 0; clinical score 0) exhibit characteristics within normal limits, or minimal to mild spongiosis in relation to the gross pathology observed at later time points. Immunohistochemistry stains Iba1 and GFAP were used to detect microglia and astrocytes, respectively, and in situ hybridisation (ISH) was used to detect viral RNA, evident by day 3–4 post-challenge (clinical score 2). A substantial increase in both microglia and astrocytes is evident by day 5–6 post-challenge (clinical score 5–7), indicative of infection/trauma. Spongiosis (arrows), neuronal cell death (morphologically compatible with apoptosis) (*), and the presence of VEEV RNA is most severe/marked by day 5–6 post-challenge.

**Figure 4 viruses-15-01307-f004:**
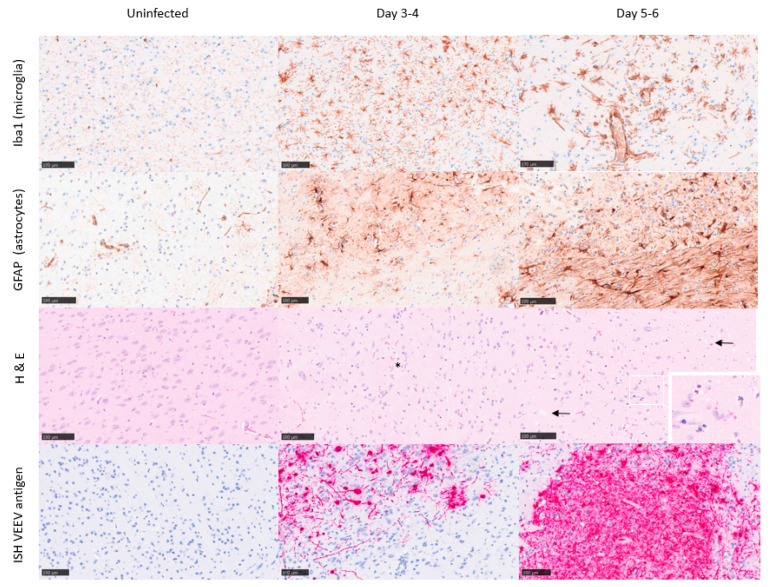
Representative images from the mid-brain/thalamus of Balb/c mice infected with VEEV TrD subcutaneously. Uninfected controls (day 0; clinical score 0) exhibit characteristics within normal limits, or minimal spongiosis in relation to the gross pathology observed at later time points. Immunohistochemistry stains Iba1 and GFAP were used to detect microglia and astrocytes, respectively, and in situ hybridisation was used to detect VEEV RNA, evident by day 3–4 post-challenge (clinical score 1–2). A substantial increase in both microglia and astrocytes is evident by day 5–6 post-challenge (clinical score 5–7), indicative of infection/trauma. Spongiosis (arrows), neuronal cell death (*), and the presence of VEEV RNA are most severe/marked by days 5–6 post-challenge.

**Figure 5 viruses-15-01307-f005:**
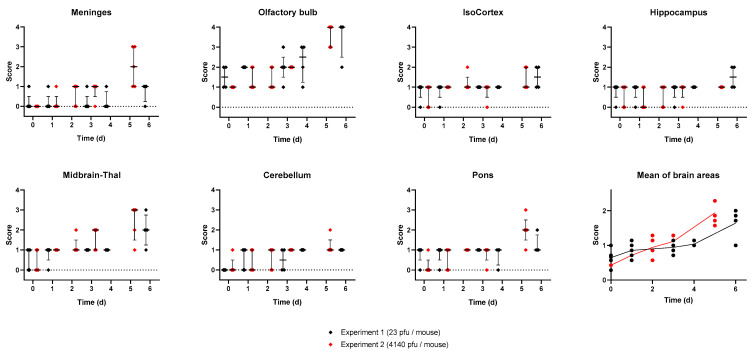
Blinded pathology scores assessing the health of various compartments of the brain from two experiments where Balb/c mice were infected with either 23 pfu (black) or 4140 pfu (red) of VEEV TrD via the subcutaneous route. Each point represents data from a single mouse, the line is the median of data, and the error bars are the interquartile range. The mean score across the compartments was also calculated and is shown last.

**Figure 6 viruses-15-01307-f006:**
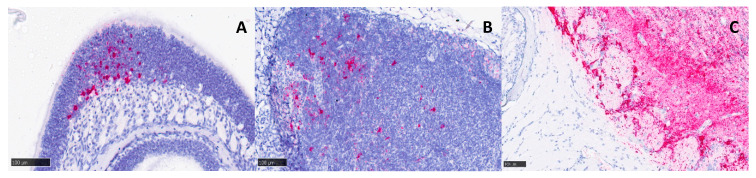
Representative images from tissues of heads of Balb/c mice infected with VEEV TrD subcutaneously and stained for viral RNA using in situ hybridisation. Despite an absence of overt histopathological changes, low levels of viral RNA were detected in the nasal cavity (**A**) and the mandibular lymph nodes (**B**) at 3+ days post-infection, with a clinical score of 2. A section across the cribriform plate (**C**) reveals an absence of viral RNA in the nerves from the nasal cavity to the olfactory bulb, but it was present within blood vessels of the olfactory bulb at 5 days post-infection (clinical score of 7).

**Figure 7 viruses-15-01307-f007:**
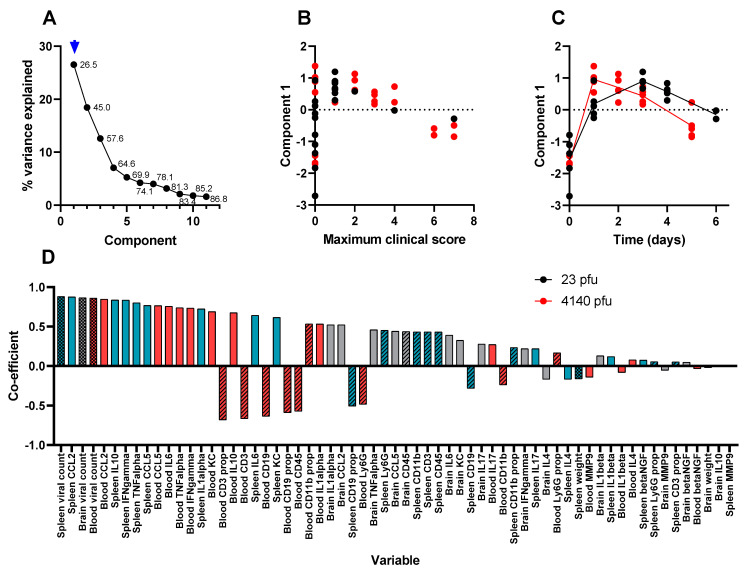
The first component in a PCA of time course data from Balb/c mice infected with (23 pfu) or (4140 pfu) VEEV TrD via the subcutaneous route. Animals were culled at time points, and multiple measurements were taken. Panel (**A**) shows the scree plot indicating the proportion of the dataset (including the data from all parameters) that can be described by each component (an amalgam of some of each parameter) calculated by the analysis. These data are shown either as a line (the proportion explained by the component) or as a data label (the cumulative proportion). An arrow has been added to indicate which component is characterised further in this figure (i.e., component 1 in this figure). Panel (**B**) shows the regression-derived value for this component for each mouse, relative to the maximum clinical score prior to cull. Each data point is from a single mouse. Panel (**C**) shows the regression-derived value for this component, for each mouse, relative to time post-challenge at point of cull. Each data point is from a single mouse with a line added to indicate the mean for each cull point. Panel (**D**) shows the coefficients, in order of absolute scale, of each variable that contributes to this component. Measurements of viral titre/body weight, cytokines, and flow cytometry are individually assigned in blood (red), spleen (blue), and brain (grey). These measurements are further divided by viral titre and animal weight (checkerboard), flow cytometry (crosshatched), and cytokine (plain).

**Figure 8 viruses-15-01307-f008:**
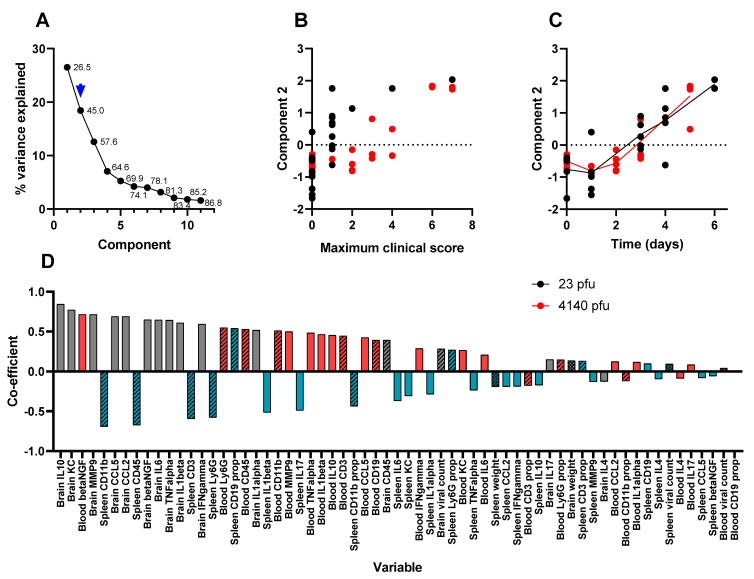
The second component in a PCA of time course data from Balb/c mice infected with (23 pfu) or (4140 pfu) VEEV TrD via the subcutaneous route. Animals were culled at time points, and multiple measurements were taken. Panel (**A**) shows the scree plot indicating the proportion of the dataset (including the data from all parameters) that can be described by each component (an amalgam of some of each parameter) calculated by the analysis. These data are shown either as a line (the proportion explained by the component) or as a data label (the cumulative proportion). An arrow has been added to indicate which component is characterised further in this figure (i.e., component 2 in this figure). Panel (**B**) shows the regression-derived value for this component for each mouse, relative to the maximum clinical score prior to cull. Each data point is from a single mouse. Panel (**C**) shows the regression-derived value for this component, for each mouse, relative to time post-challenge at point of cull. Each data point is from a single mouse with a line added to indicate the mean for each cull point. Panel (**D**) shows the coefficients, in order of absolute scale, of each variable that contributes to this component. Measurements of viral titre/body weight, cytokines, and flow cytometry are individually assigned in blood (red), spleen (blue), and brain (grey). These measurements are further divided by viral titre and animal weight (checkerboard), flow cytometry (crosshatched), and cytokine (plain).

**Figure 9 viruses-15-01307-f009:**
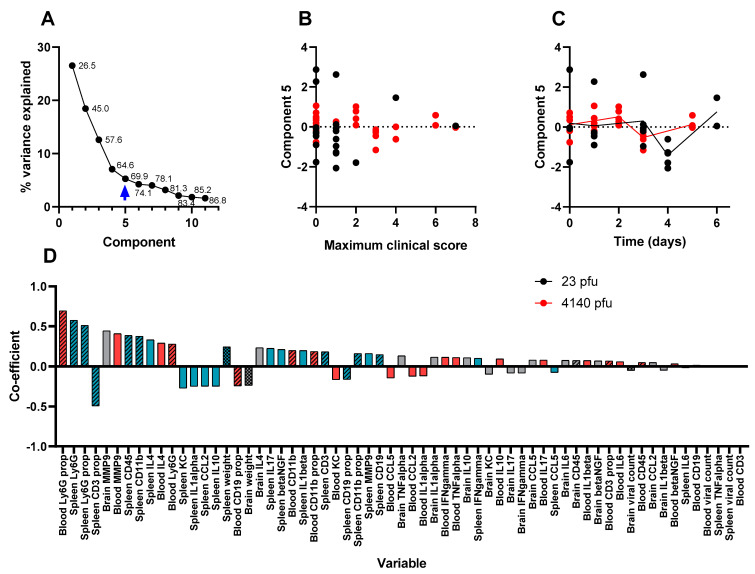
The fifth component in a PCA of time course data from Balb/c mice infected with (23 pfu) or (4140 pfu) VEEV TrD via the subcutaneous route. Animals were culled at time points, and multiple measurements were taken. Panel (**A**) shows the scree plot indicating the proportion of the dataset (including the data from all parameters) that can be described by each component (an amalgam of some of each parameter) calculated by the analysis. These data are shown either as a line (the proportion explained by the component) or as a data label (the cumulative proportion). An arrow has been added to indicate which component is characterised further in this figure (i.e., component 5 in this figure). Panel (**B**) shows the regression-derived value for this component for each mouse, relative to the maximum clinical score prior to cull. Each data point is from a single mouse. Panel (**C**) shows the regression-derived value for this component, for each mouse, relative to time post-challenge at point of cull. Each data point is from a single mouse with a line added to indicate the mean for each cull point. Panel (**D**) shows the coefficients, in order of absolute scale, of each variable that contributes to this component. Measurements of viral titre/body weight, cytokines, and flow cytometry are individually assigned in blood (red), spleen (blue), and brain (grey). These measurements are further divided by viral titre and animal weight (checkerboard), flow cytometry (crosshatched), and cytokine (plain).

**Figure 10 viruses-15-01307-f010:**
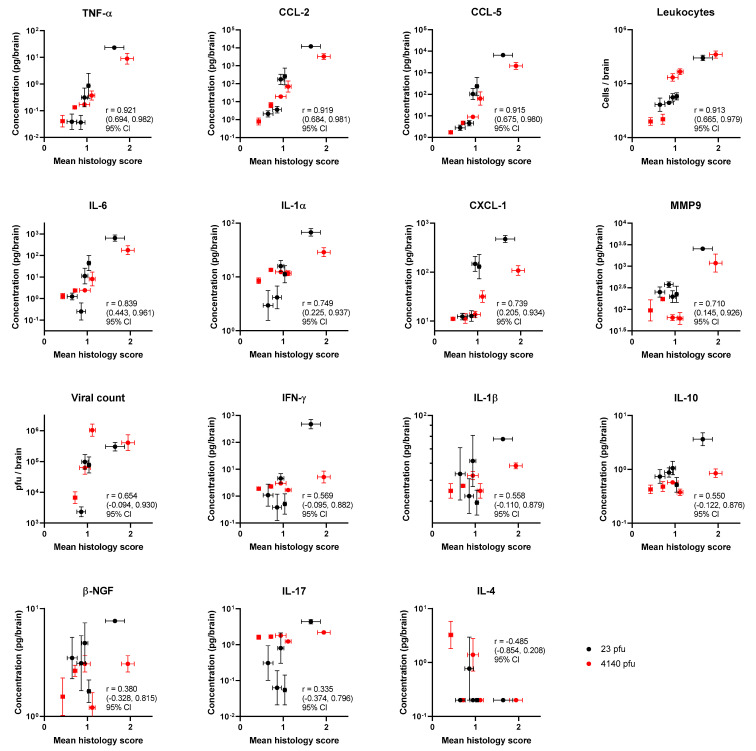
Mean blinded pathology scores assessing the health of various compartments of the brain in relation to the immune/infection markers from two experiments where Balb/c mice were infected with either 23 pfu (red) or 4140 pfu (black) of VEEV TrD via the subcutaneous route. Each data point is the mean pathology score of 5 mice (±SEM) and the geometric mean of cytokine levels (±geometric SEM). The immune markers are arrayed by strength of Pearson’s correlation.

**Table 1 viruses-15-01307-t001:** Quantitative score assigned to tissue sections from the brains and nasal compartments of Balb/c mice infected with VEEV TrD.

Pathology Score	Histopathological Lesions in the Brain	Perivascular Cuffing within Meninges	Histopathology in the Nasal Cavity (inc. Mucosa)
**0**	Within normal limits	Within normal limits	Within normal limits
**1**	Minimal spongiosis	Minimal	Minimal
**2**	Mild spongiosis and minimal to mild increase in neuropil cellularity (glia)	Mild	Mild
**3**	Moderate spongiosis and mild neuronal death (as observed within neuronal soma) Moderate increase in neuropil cellularity (glia)	Moderate	Moderate
**4**	Moderate to severe spongiosis and neuronal death (as observed within neuronal soma) Moderate increase in neuropil cellularity (glia)	Marked/severe	Marked/severe necrosis of mucosa and presence of exudate within lumen

**Table 2 viruses-15-01307-t002:** Summary of the findings and experimental interpretations of the five principal components determined using the principal component analysis technique.

Principal Component	Percentage of the Data Related to Component	Main Associated Components	Correlation to Clinical Signs of Disease?	Interpretation
**1**	26.5%	Th1-biased inflammation markers in the blood and spleen, varying considerably between time points Viral titres in blood and spleen, rapid increase with slow decline	No	Systemic disease, with rapid conventional host response
**2**	18.5%	Th1-biased inflammation markers in the brain, varying considerably between time points Viral titres in brain, rapid increase over time	Yes	A strong association with clinical signs of disease (typically neurological), likely representative of the encephalitis typical of lethal infection
**3**	12.6%	Th2- and Th17-biased inflammation markers in all sample types did not differ between time points	No	Inter-experimental variation in flow cytometry analysis
**4**	7.0%	Cell counts in the blood and spleen did not vary between time points	No	Inter-experimental variation in luminex analysis
**5**	5.3%	Leukocyte counts in the blood, transient decline at the midway point of sampling	No	Leukopenia

## Data Availability

The data file used to generate all analyses and figures are within Appendix A.

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
