# Peer review of "Tumour Necrosis Factor-α, Chemokines, and Leukocyte Infiltrate Are Biomarkers for Pathology in the Brains of Venezuelan Equine Encephalitis (VEEV)-Infected Mice"

_viruses, 2023, doi:10.3390/v15061307_

Round 1
Reviewer 1 Report
This research manuscript seeks to characterize VEEV-induced inflammation in a mouse model in order to identify inflammatory biomarkers that contribute to pathology in the brain.
Balb/c mice were subcutaneously infected with VEEV TrD at a low or high dose. Minor clinical symptoms were observed for 1-2 days before the disease progressed to severe clinical symptoms, which is consistent with reported literature. Highest quantities of viral RNA were found in the olfactory bulb and midbrain/thalamus which corresponded to the areas of highest degree of pathology.
1) For the pathological images of the olfactory bulb, midbrain/thalamus, and whole head (Fig 4, 5, & 6), please include information regarding the selected animal's corresponding clinical score. A table outlining the clinical score and its association to pathology would be helpful.
2) Please add a figure summarizing the results of the Principle Component Analysis and the 5 principal components
Author Response
We thank the reviewer for their helpful comments. The clinical score data has been added to (Fig 4, 5, & 6). We spent some time thinking about what a “figure summarizing the results of the Principle Component Analysis” might look like. We think that a table would be best used for this purpose and one has been added.
Reviewer 2 Report
An overall well-written paper of a good scientific standard.
See next entry for suggested corrections.
Tumour Necrosis Factor-α, Chemokines and Leukocye Infiltrate are Biomarkers for Pathology in the Brains of Venezuelian Equine Encephalitis (VEEV) Infected Mice
Herewith some suggested corrections
Responses marked by *. Comments denoted in italics
Line 2 to 4: use appropriate capitalisation for heading
* Tumour Necrosis Factor-α, Chemokines and Leukocye Infiltrate are Biomarkers for Pathology in the Brains of Venezuelan Equine Encephalitis (VEEV) Infected Mice
Line 39: …Investigational New Drug (IND) vaccine..
*…investigational new drug (IND) vaccine
Capitalise if referring to the name of the programme
Line 42: .. Phase I clinical trials…
* .. phase I clinical trials…
Line 62: Sindbis Virus
*Sindbis virus
Line 64: ..of Tumour Necrosis Factor… Intereron….Interleukin…
* .of tumour necrosis factor… intereron….interleukin…
Line 66 : TNF Receptor
*TNF receptor
Line 67: Nitric Oxide-Synthesis
*..nitric oxide synthase
Line 70: Blood Brain Barrier
*..blood-brain barrier
Line 73; ….Severe Combined Immuno-Deficient …
* ..severe combined immunodeficiency ….
Line 199: …Foetal Bovine Serum…...
*..foetal bovine serum..
Line 229: …and the ‘polymer refine detection kit..’
*… and the Bond Polymer Refine Detection kit..
Line 238: Rabbit polyclonal
..rabbit polyclonal..
Line 255 and elsewhere for the next several pages: Error! Reference source not found
*Correct these reference errors
Line 325:… and regards to Kaplan-Meier estimate..
..and as regards to the Kaplan-Meier estimate..
Line 406: ….used to analysis these data
* used to analyse these data
Line 569 & 570: ..Tocilizumab..
*..tocilizumab ( generic names not capitalised)
Line 575, 577: …as Etanercept and Anakinra…
* as etanercept and anakinra …. ( generic names not capitalised)
Line 585 : ..are the Non-Steroidal Anti-Inflammatory Drugs
* are the non-steroidal anti-inflammatory drugs ..
Line593:.. ..Celecoxib…..
*. ..celecoxib….. (generic names not capitalised)
Line 595: .. was Baricitinib…
*.. was baricitinib…(generic names not capitalised)
Line 596:…Janus Kinases
*..Janus kinases…
References
Use capitals appropriately for journal names :
eg 653 : Journal of virology
*Journal of Virology
and elsewhere, see reference numbers 12, 13, 17, 26, 28, 29, 30, 31, 36, 37, 45, 51, 54, 62.
Author Response
We thank the reviewer for their helpful comments. We have addressed all of these errors in the upload version.
Reviewer 3 Report
This manuscript investigates infection of Balb/c mice with VEEV, an emergin pathogen with potencial for biowarefare. This is a tour the force analysis of the pathology and immunological responses to VEEV infections. The authors include 45 mice for each group and perform a extensive analysis of inflammatory biomarkers, virus replication and pathology. Most impressively, they analyse the comprehensive set of data using an usuppervised learning technique to conclude the relevance of certain cytokines responses that correlate with inflammation in the brain. Based on this rigorous analysis the authors discuss an entyire set of new approaches to controling adverse effect from the infection that they intent to pursue in future studies. Overall this is an excellent study that will serve as a model for many future strudies. It will be of great value to a broad audience aroun the world.
One minor comment the authors should include one reference:
CDC.Venezuelan equine encephalitis - Colombia, 1995.MMWR Morb Mortal Wkly Rep. 1995;44(39):721-724.
Author Response
We thank the reviewer for their helpful comments. We have added the reference.